# DDRprog: A CLEVR Differentiable Dynamic Reasoning Programmer

## Abstract

We present a generic dynamic architecture that employs a problem specific differentiable forking mechanism to leverage discrete logical information about the problem data structure. We adapt and apply our model to CLEVR Visual Question Answering, giving rise to the DDRprog architecture; compared to previous approaches, our model achieves higher accuracy in half as many epochs with five times fewer learnable parameters. Our model directly models underlying question logic using a recurrent controller that jointly predicts and executes functional neural modules; it explicitly forks subprocesses to handle logical branching. While FiLM and other competitive models are static architectures with less supervision, we argue that inclusion of program labels enables learning of higher level logical operations – our architecture achieves particularly high performance on questions requiring counting and integer comparison. We further demonstrate the generality of our approach though DDRstack – an application of our method to reverse Polish notation expression evaluation in which the inclusion of a stack assumption allows our approach to generalize to long expressions, significantly outperforming an LSTM with ten times as many learnable parameters.

## 1 Introduction and Related Works

Deep learning is inherently data driven – visual question answering, scene recognition, language modeling, speech recognition, translation, and other supervised tasks can be expressed as: given input $x$, predict output $y$. The field has attempted to model different underlying data structures with neural architectures, but core convolutional and recurrent building blocks were designed with only general notions of spatial and temporal locality. In some cases, additional information about the problem can be expressed simply as an additional loss, but when hard logical assumptions are present, it is nonobvious how to do so in a manner compatible with backpropagation.

Discrete logic is a fundamental component of human visual reasoning, but there is no dominant approach to incorporating such structural information into deep learning models. For particular data structures and settings, there has been some success. However, in prior work additional structure must either be learned implicitly and without additional annotations or is available at both train and test time. For example, StackRNN (Joulin & Mikolov (2015)) allows recurrent architectures to push and pop from a stack. While this approach works well without explicit stack trace supervision, implicit learning only goes so far: the hardest task it was tested on is binary addition. Approaches such as recursive NN (Socher et al. (2011)) and TreeRNN (Tai et al. (2015)) allow inclusion of explicit tree structures available during both training and testing, but neither can be used when additional supervision is available only at training time. We consider this the most general problem because it is not feasible to obtain good results without any additional supervision if the problem is sufficiently difficult.

Our objective is to develop a general framework for differentiable, discrete reasoning over data structures, including as stacks and trees. Our approach is flexible to differing degrees of supervision and demonstrates improved results when structural assumptions are available at test time. We are less concerned with the no-supervision case because of limitations in scalability, as demonstrated by the scope of StackRNN.

We present our framework in the context of two broader architectures: Neural Module Networks (NMN, Andreas et al. (2015)) and Neural Programmer-Interpreters (NPI, Reed & de Freitas (2015)).

The original NMN allows per-example dynamic architectures assembled from a set of smaller models; it was concurrently adapted in N2NMN (Hu et al. (2017)) and IEP as the basis of the first visual question answering (VQA) architectures successful on CLEVR (Johnson et al. (2016)). The NPI work allows networks to execute programs by directly maximizing the probability of a successful execution trace. In the present work, we present two applications of our framework, which is a superset of both approaches. The first is our CLEVR architecture, which introduces two novel behaviors. It interleaves program prediction and program execution by using the output of each module to predict the next module; this is an important addition because it improves the differentiability of the model. For IEP/N2NMN, the discrete program in the middle of the model breaks the gradient flow. For our model, although the selection of modules is still a discrete non-differentiable choice, it is influenced by the loss gradient: the visual state gives a gradient pathway learnable through the question answer loss. The second contribution of this architecture is a novel differentiable forking mechanism that enables our network to process logical tree structures through interaction with a stack of saved states. This allows our model to perform a broad range of logical operations; DDRstack is the first architecture to obtain consistently strong performance across all CLEVR subtasks.

We briefly discuss our rationale for evaluation on CLEVR as well as prior work on the task. Though CLEVR is too easy with or without program supervision, it is the best-available proxy task for high-level reasoning. Its scale, diverse logical subtask categories, and program annotations make the dataset the best current option for designing discrete visual reasoning systems. By effectively leveraging the additional program annotations, we improve over the previous state-of-the-art with a much smaller model – on the important Count and Compare Integer subtasks, we improve from 94.5 to 96.5 percent and 93.8 to 98.4 percent, respectively. However, our objective is neither the last couple percentage points of accuracy on this task nor to decrease supervision, but to motivate more complex tasks over knowledge graphs. We expect that it is possible to improve accuracy on CLEVR with a static architecture using less supervision. This is largely unrelated to the objective of our work – we view CLEVR as a good first step towards increased supervision for the learning of complex logic. Human-level general visual reasoning from scratch is less reasonable than from expressively annotated data: we consider improving and generalizing the ability of architectures to better leverage additional supervision to be the most likely means to this end.

Prior work on CLEVR is largely categorized by dynamic and static approaches. IEP (Johnson et al. (2017)) and N2NMN both generalized the original neural module networks architecture and used the functional annotations in CLEVR to predict a static program which is then assembled into a tree of discrete modules and executed. IEP further demonstrated success when program annotations are available for only a few percent of questions. These are most similar to our approach; we focus largely upon comparison to IEP, which performs significantly better. RN (Santoro et al. (2017)) and FiLM (Perez et al. (2017b)), the latter being the direct successor of CBN (Perez et al. (2017a)) are both static architectures which incorporate some form of implicit reasoning module in order to achieve high performance without program annotations. In contrast, our architecture uses program annotations to explicitly model the underlying question structure and jointly executes the corresponding functional representation. As a result, our architecture performs comparably on questions requiring only a sequence of filtering operations, but it performs significantly better on questions requiring higher level operations such as counting and numerical comparison.

We present DDRstack as a second application of our framework and introduce a reverse Polish notation (RPN) expression evaluation task. The task is solvable by leveraging the stack structure of expression evaluation, but extremely difficult without additional supervision: a much larger LSTM baseline fails to attain any generalization on the task. We therefore use RPN as additional motivation for our framework, which introduces a simple mechanism for differentiably incorporating the relevant stack structure. Despite major quantitative differences from CLEVR VQA, the RPN task is structurally similar. In the former, questions seen at training time contain direct programmatic representations well modeled by a set of discrete logical operations and a stack requiring at most one recursive call. The latter is an extreme case with deep recursion requiring a full stack representation, but this stack structure is also available at test time.

In summary: the DDR framework combines the discrete modular behavior of NMN and IEP with an NPI inspired forking behavior to leverage structural information about the input data. Our approach resolves common differentibility issues and is easily adapted to the specific of each problem: we achieve a moderate improvement over previous state-of-the-art on CLEVR and succeed on RPN where a much larger baseline LSTM fails to attain generalization.

## 2 DATASETS

### 2.1 CLEVR

CLEVR is a synthetic VQA dataset that encourages approaches capable of discrete reasoning through its inclusion of functional program annotations that model the logic of each question. The dataset consists of 100k images and 1 million question/answer pairs. Over 850k of these questions are unique. Images are high quality 3D Blender (Blender (2017)) renders of scenes containing geometric objects of various shapes, sizes, colors, textures, and materials. Thus the dataset is quite realistic despite being synthetic. Furthermore, the authors ran comprehensive tests to avoid exploitable biases in the data. As the objects are geometric figures, no external knowledge of natural images is required, as in earlier VQA datasets. Most importantly, CLEVR provides an expressive program representation of each question. For example, "How many red spheres are there?" is represented as [*filter_red, filter_sphere, count*]. Some questions require nonlinear program structure, such as "How many objects are red or spheres", which is represented by a tree with two branches [*filter_red*] and [*filter_sphere*] followed by a binary [*union*] operation and a final [*count*]. We include additional examples in the appendix. We are unaware of any dataset with comparable properties of this size and complexity.

By raw accuracy alone, CLEVR is effectively solved. Neural architectures have already far surpassed human accuracy on the task, both with and without using program annotations at train time. One perspective is that this should motivate a return to the natural image setting without programs or with transfer learning from CLEVR. In contrast, we believe the rapid recent progress on CLEVR motivates more complex synthetic tasks – perhaps involving harder logical inference over general knowledge graphs. It is not currently obvious what form a visual Turing test should take, nor is it clear what should comprise the train set for such a task: this will likely require much iteration and experimentation. On this front, the synthetic setting is unmatched: making even slight changes to a natural image dataset often involves a length additional data collection task compared to a programmatic change in the synthetic case.

### 2.2 RPN

We introduce the reverse Polish notation (RPN) expression evaluation dataset as a motivation for additional supervision in higher level learning tasks. The specific problem form we consider is [NUM]*($n$+1)-[OP]*$n$, that is, $n + 1$ numbers followed by $n$ operations. For example, "2 3 4 + *" evaluates to 14. This simplifies the problem by eliminating consideration for order of operations. Thus the task is: given a sequence of tokens corresponding to a valid expression in reverse Polish notation, evaluate the expression and produce a single real valued answer.

This may seem like a simple task; it is not. For large $n$, expressions behave somewhat like a hash function. Small changes in the input can cause wild variations in the output – we found the problem intractable in general. Our objective is to make stronger structural assumptions about the problem and create an architecture to leverage them. For this reason, our framework is incomparable to StackRNN, which attempts to learn a stack structure implicitly but is unable to incorporate additional supervision when the problem is likely too difficult to solve otherwise. We therefore modify the problem as such: instead of producing only the final expression evaluation, produce the sequence of answers to all $n$ intermediate expressions in the answer labels. For the example "2 3 4 + *", the expected output would be [7, 14] because 3+4=7 and 2*7=14. We further assume the stack structure of the problem is available to the architecture should it be capable of taking advantage of such information. The problem is still sufficiently complex – note that to the model, $\{1, 3, 4, +, *\}$ would all be meaningless tokens: it must learn both the NUM and the OP tokens.

The dataset consists of 100k train, 5k validation, and 20k test expression with $n = 10$ – that is, 11 numbers followed by 10 operations. We also provide a 20k expression generalization set with $n = 30$. The label for each question contains the $n$ solutions to each intermediate operation. During data generation, we sample NUM and OP tokens uniformly, reject expressions including division by zero, and also omit expressions that evaluate to over 100 in magnitude. The NUM tokens correspond to 0, 0.1, ..., 0.9 and the OP tokens correspond to +, -, *, /; however, architectures are not privy to this information.

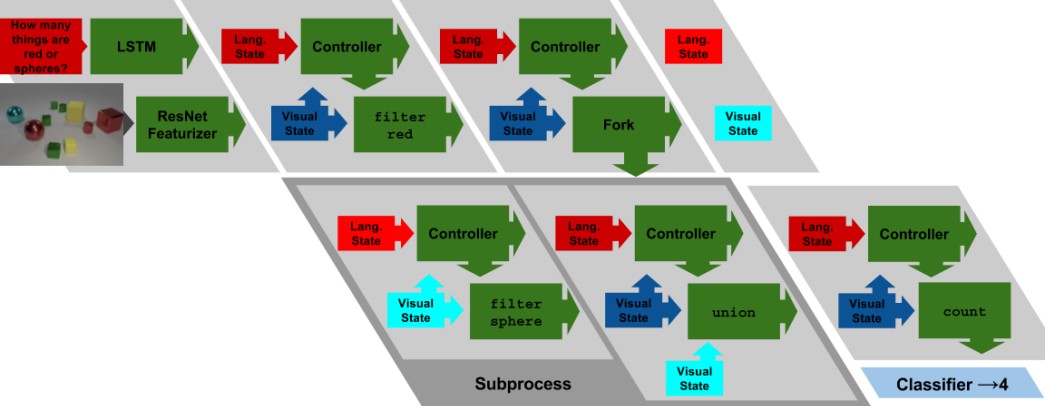

Figure 1: Visualization of the DDRprog architecture. This configuration answers "How many things are red or spheres?" by predicting $[filter\_red, fork, filter\_sphere, union, count]$

# 3 DDR Architecture

The purpose of the DDR framework is to naturally incorporate structured information into a neural reasoning architecture. This is important when the given problem is hard without additional supervision and allows the model to perform discrete, complex reasoning. Our framework addresses the difficulty of combining discrete logic with clean, differentiable training and is capable of interfacing with a broad range of data structures. Our framework is a clean fusion of two broad design patterns. Like IEP, we maintain a set of problem-specific neural modules to allow our model to learn relevant program primitives. Like NPI, we interleave program prediction with program execution, differentiably learning modules when module arrangement is not known at test time.

This is much more general compared to either IEP/NMN or NPI independently, and the particular mechanism for combining them is non-trivial differentiable forking operation. IEP alone lacks the ability to examine the output of intermediate operations. The relevance of this is particularly clear in the CLEVR setting. The NPI architecture can learn sequences of functions, but lacks the ability to learn the functions themselves. Our approach responds flexibly to the problem supervision: in VQA, modules are known only at train time. At each timestep, the controller therefore produces an index corresponding to a neural module, which is then executed. On the RPN task, the problem structure is also known at test time; the controller is therefore deterministic and directly executes the correct module. We refer to our VQA and RPN architecture adaptations as DDRprog and DDRstack, respectively; details are provided below.

## 3.1 CLEVR Visual Question Answering: DDRprog

DDRprog is a direct adaptation of our framework to CLEVR, requiring only some standard encoders to handle the mixed language and visual data. We provide pseudocode in Algorithm 1, a visual representation in Figure 1, and subnetwork details in Table 2 (see Appendix).

The input data $x$ for each sample is a (image, question, program) triple; the label $y$ is a (answer, program) pair. The program only available at train time, thus our model must learn to predict it.

The network first applies standard LSTM and ResNet (He et al. (2015)) encoders to the question/image, producing language and visual states. The ResNet encoder is unchanged from FiLM/IEP.

Both the language and visual states are passed to the controller. We use a recurrent highway network (RHN) (Zilly et al. (2016)) as recommended by Suarez (2017) instead of an LSTM(Hochreiter & Schmidhuber (1997)) – both accept flat inputs. As the visual state contains convolutional maps, we flatten it with a standard classifier.

At each time step, the controller outputs a standard softmax classification prediction, which is interpreted as an index over the set of learnable neural modules. These are smaller, slightly modified variants of the modules used in IEP. The selected module is executed on the visual state;

---

**Algorithm 1** DDRprog. Note that **CNN** produces a flattened output and **Controller** also performs a projection and argmax over program scores to produce $programPrediction$

---

$img, question \leftarrow x$
$stack \leftarrow$ **Stack()**
$img, imgCopy \leftarrow$ **ResNetFeaturizer(**$img$**)**
$langState \leftarrow$ **LSTM(**$question$**)**
**for** $i = 1...MaxProgramLength$ **do**
    $visualState \leftarrow$ **CNN(**$img$**)**
    $programPrediction \leftarrow$ **Controller(**$visualState, langState$**)**
    $cell \leftarrow$ **Cells[**$programPrediction$**]**
    **if** $cell$ **is** $Fork$ **then**
        $stack.$**push(**$cell(img, imgCopy)$**)**
    **else if** $cell$ **is** $Binary$ **then**
        $img \leftarrow cell(stack.$**pop()**$, img)$
    **else**
        $img \leftarrow cell(img)$
    **end if**
**end for**
**return Classifier(**$img$**)**

---

the visual state is set to the output. The module prediction at the final timestep is followed by a small classifier network, which uses the IEP classifier. This architecture introduces a significant advantage over IEP: as modules are predicted and executed one at a time instead of being compiled into a static program, our model can observe the result of intermediate function operations – these have meaning as filtering and counting operations on CLEVR.

We now motivate our differentiable forking mechanism. As presented thus far, our approach is sufficient on the subset of CLEVR programs that do not contain comparison operations and are effectively linear – indeed, we observe a large performance increase over IEP on this subset of CLEVR. However, some CLEVR questions contain a logical branching operation (e.g. *are there more of ... than ... ?*) and cannot be answered by structurally linear programs. In general, programs can take the form of expressive trees, but CLEVR programs contain at most two branches. Adding a differentiable forking mechanism handles the general case without modification from the CLEVR architecture. Upon encountering a program branch, our architecture pushes the current language and visual states to a stack and forks a subprocess. This subprocess is effectively a copy of the main network that maintains its own states. It takes as input the language state and the initial and current visual states. Effectively, a different copy of the network with its own state processes each branch of the program. Upon processing the last operation in the branch, a binary cell is applied to the subprocess state outputs and the main process states (popped from the stack), merging them as shown in Figure 1. Our architecture is likely to generalize even past the setting of tree processing, as we could replace the stack with a priority queue or any other data structure pertinent to the problem.

Finally, a technical note for reproducibility: the fork module must differ from a standard unary module, as it is necessary to pass the original ResNet features (e.g. the initial visual state) to the subprocess in addition to the current visual state. Consider the question: "Is the red thing larger than the blue thing?" In this case, the main network filters by red; it is impossible to recover the blue objects in the subprocess given a red filtered image. We found that it is insufficient to pass only the original images to the subprocess, as the controller is small and has difficulty tracking the current branch. We therefore use a variant of the binary module architecture that merges the original ResNet features with the current visual state (see Algorithm 1). As the fork module is shared across all branch patterns, it is larger than the other binary modules and also one layer deeper – refer to the appendix for full architecture details on each layer.

### 3.2 EXPRESSIONS IN REVERSE POLISH NOTATION: DDRSTACK

The DDRstack architecture applies our general framework to the increased supervision setting of the RPN task – module arrangement is a fixed expression parse tree. One natural view of the task is: given a parse tree structure, simultaneously socket and refine the learnable NUM and OP nodes.

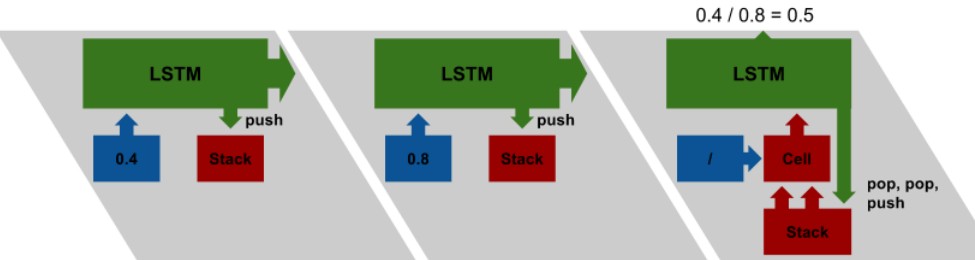

Figure 2: Visualization of the DDRstack architecture with $n = 1$. This particular configuration evaluates the [NUM][NUM][OP] formatted expression [0.4, 0.8, /], which is 0.4/0.8=0.5. NUM tokens are embedded before being passed to the LSTM. OP tokens are used as an index to select the corresponding cell. LSTM predictions at each OP token are used to predict intermediate losses (there is only one for $n = 1$).

Our model consists of an LSTM controller and a set of 4 learnable binary modules – one per OP – as well as an explicit stack. DDRstack processes one token at a time; similar to unary/binary modules in IEP, NUM and OP tokens are processed differently:

NUM: Our model embeds the token and passes it to the LSTM. It then pushes the result to the stack.

OP: Our model pops twice, calls the OP specific binary cell, and then passes the results to the LSTM. It then pushes the result to the stack. The binary cell is a simple concatenation of the arguments followed by a single fully connected layer with no nonlinearity.

DDRstack can be viewed as a neural analog to standard analytical RPN expression evaluation algorithm where the values of the NUM and OP tokens are unknown. We provide high level pseudocode for the model in Algorithm 2 and a visual representation in Figure 2.

---

**Algorithm 2** DDRstack

$tokens \leftarrow x$
$stack \leftarrow$ **Stack()**
$state \leftarrow RandomLSTMInitialization$
**for all** $tok$ **in** $tokens$ **do**
    **if** $tok$ **is a NUM then**
        $out \leftarrow$ **Embed(**$tok$**)**
    **else if** $tok$ **is a OP then**
        $arg2 \leftarrow stack$**.pop()**
        $arg1 \leftarrow stack$**.pop()**
        $out \leftarrow$ **Cells[**$tok$**](**$arg1, arg2$**)**
    **end if**
    $out, state \leftarrow$ **LSTMCell(**$out, state$**)**
    $stack$**.push(**$out$**)**
**end for**
**return Projection(**$out$**)**

---

We train a baseline vanilla LSTM supervised with the intermediate solutions in the last $n$ timesteps. DDRstack uses the same baseline LSTM as its core controller, but includes the aforementioned stack behavior. For both models, predictions are made in the last $n$ timesteps (Algorithm 2 shows only the final return).

## 4 DDRPROG: CLEVR VQA EXPERIMENTS AND DISCUSSION

### 4.1 EXPERIMENTS

We train our model with Adam (Kingma & Ba (2014)) on the full CLEVR dataset, including all program annotations. The hyperparameters are detailed in Table 3 (see Appendix) – these, along with the structure of the subnetworks, are minimally configured from defaults, excepting learning

| Model | Parameters | Epochs | Exist | Count | Compare Integer | Query | Compare | Overall |
|---|---|---|---|---|---|---|---|---|
| Q-type mode | - | - | 50.2 | 34.6 | 51.1 | 36.9 | 51.2 | 42.1 |
| LSTM | - | - | 61.8 | 42.5 | 70.0 | 36.5 | 51.1 | 47.0 |
| CNN+LSTM | - | - | 68.2 | 47.8 | 70.1 | 48.9 | 54.6 | 54.3 |
| CNN+LSTM+SA | - | - | 68.4 | 57.5 | 67.7 | 87.7 | 52.0 | 69.8 |
| CNN+LSTM+SA+MLP | - | - | 77.9 | 59.7 | 75.1 | 80.9 | 70.8 | 73.2 |
| Human | - | - | 96.6 | 86.7 | 86.4 | 94.9 | 96.0 | 92.6 |
| End-to-End NMN* | - | - | 85.7 | 68.5 | 84.9 | 89.9 | 88.7 | 83.7 |
| IEP* | 41M | 12 | 97.1 | 92.7 | **98.7** | 98.1 | 98.8 | 96.9 |
| DDRprog* | 9M | 52 | 98.8 | **96.5** | 98.4 | 99.1 | **99.0** | **98.3** |
| RN | 500k | 1000 | 97.8 | 90.1 | 93.6 | 97.9 | 97.1 | 95.5 |
| FiLM/CBN | >50M | 80 | **99.2** | 94.5 | 93.8 | **99.2** | **99.0** | 97.6 |

Table 1: Accuracy on all CLEVR question types for baselines and competitive models. The Human baseline is from the original CLEVR work. * denotes additional program supervision. SA refers to stacked spatial attention Yang et al. (2015)

.

rate and rough network size. The network overall has 9M parameters. We exclude the ResNet feature extractor from all calculations because it is also present in the best FiLM model. Their work further demonstrated it is fairly straightforward to replace it with a from-scratch feature extractor with minimal loss in accuracy.

We pass the ground truth program labels to the model during training. Critically, the program labels are only used on the validation set for the purpose of model selection, and our final accuracy is obtained by rerunning the validation set without the ground truth programs. We train on a single GTX 1080 TI and, after 35 epochs, our model matches the previous state-of-the-art accuracy of 97.7 percent. We continue training until the 52nd epoch, dropping the learning rate to 1e-5 for the last few epochs to ensure convergence, and obtain 98.3 percent accuracy. The model predicts program cells with 99.98 percent accuracy.

## 4.2 DISCUSSION

Several models have far exceeded human accuracy on CLEVR – the task remains important for two reasons. First, though CLEVR is large and yields consistent performance across runs, different models exhibit significantly different performance across question types. Where every competitive previous work exhibits curiously poor performance on at least one important subtask, our architecture dramatically increases consistency across all tasks. Second, CLEVR remains the best proxy task for high-level visual reasoning because of its discrete program annotations – this is far more relevant than raw accuracy to our work, which is largely concerned with the creation of a general reasoning framework. However, we do achieve a modest improvement in raw accuracy over the previous state-of-the-art with a >5X smaller architecture.

We presently consider RN, FiLM, IEP, and our architecture as competitive models. From Table 1, no architecture has particular difficulty with Exist, Query, or Compare questions; the main differentiating factors are Count and Compare Integer. Though Compare Integer is the smallest class of questions and is therefore assigned less importance by the cross entropy loss, the IEP result suggests that this does not cause models to ignore this question type. We therefore consider Count and Compare Integer to be the hardest unary and binary tasks, respectively, and we assign most important to these question subsets in our analysis. We achieve strong performance on both subtasks and a significant increment over previous state-of-the-art on the Count subtask.

We first compare to IEP. Our model is 4x smaller than IEP (see Table 1) and resolves IEP's poor performance on the challenging Count subtask. Overall, DDRprog performs at least 2x better across all unary tasks (+1.7 percent on Exist, +3.8 percent on Count, + 1.0 percent on Query) it closely matches binary performance (+0.2 percent on Compare, -0.3 percent on Compare Integer). We believe that our model's lack of similar gains on binary task performance can be attributed to the use of a singular fork module, which is responsible for cross-communication during prediction of both branches of a binary program tree, shared across all binary modules. We have observed that this

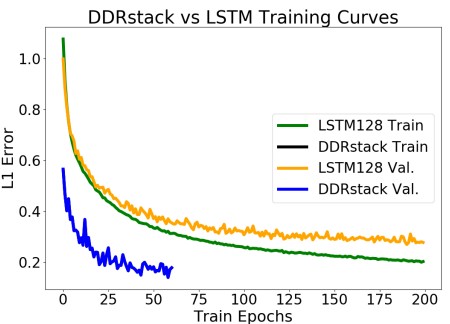 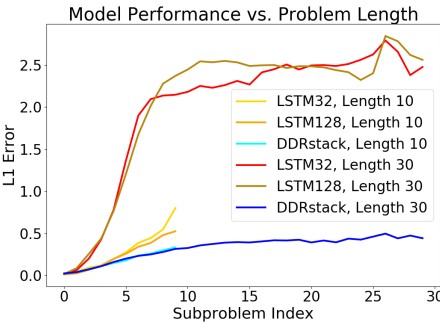

Figure 3: Left: Training curves for DDRstack (train/val overlapping, 17k parameters) and the LSTM128 baseline (255k parameters) on RPN10. Right: Generalization performance of DDRstack and the LSTM baseline to RPN30 after training on RPN10.

module is essential to obtaining competitive performance on binary tasks; it is likely suboptimal to use a large shared fork module as opposed to a separate smaller cell for each binary cell.

Our model surpasses RN in all categories of reasoning, achieving a 2.6x reduction in overall error. RN achieves impressive results for its size and lack of program labels. However, it is questionable whether the all-to-all comparison model will generalize to more logically complex questions. In particular, Count operations do not have a natural formulation as a comparison between pairs of objects, in which case our model achieves a significant 6.4 percent improvement. RN also struggles on the challenging Compare Integer subtask, where we achieve a 4.8 percent improvement. Furthermore, it is unclear how essential high epoch counts are to the model's performance. As detailed in Table 1, RN was trained in a distributed setting for 1000 epochs. Both our result and FiLM were obtained on single graphics cards and were only limited in number of epochs for practicality – FiLM had not fully converged, and our model was unregularized.

Both IEP and our model achieve a roughly 4x improvement over FiLM on Compare Integer questions (4.9 and 4.6 percent, respectively), the difference being that our model eliminates the Count deficiency and is also 4X smaller than IEP. The contrast between FiLM's Compare Integer and Exist/Query/Compare performance suggests a logical deficiency in the model – we believe it is difficult to model the more complex binary question structures using only implicit branching through batch normalization parameters. FiLM does achieve strong Compare Attribute performance, but many such questions can be more easily resolved through a sequence of purely visual manipulations. FiLM achieves 1.5x relative improvement over our architecture on Exist questions, but this is offset by our 1.5x relative improvement on Count questions.

Given proximity in overall performance, FiLM could be seen as the main competitor to our model. However, they achieve entirely different aims: DDRprog is an application of a general framework, >5X smaller, and achieves stable performance over all subtasks. FiLM is larger and suffers from a significant deficiency on the Compare Integer subtask, but it uses less supervision. As mentioned in the introduction, our model is part of a general framework that expands the ability of neural architectures to leverage discrete logical and structural information about the given problem. In contrast, FiLM is a single architecture that is likely more directly applicable to low-supervision natural image tasks.

## 5 DDRSTACK: RPN EXPERIMENTS AND DISCUSSION

### 5.1 EXPERIMENTS

For our architecture, we use hidden dimension 32 throughout the model, resulting in only 17k parameters overall. We train with Adam using learning rate 1e-3 and obtain a test L1 error of 0.17 after 63 epochs. Using the same hidden dimension in the pure LSTM baseline (9k parameters) results in test error 0.28. We overcompensate for the difference in model size by increasing the hidden dimension of the LSTM to 128 (255k parameters), resulting in an only slightly lower test error of 0.24 after nearly 3000 epochs. Figure 3 shows training curves for the LSTM baseline and DDRstack.

After training both models on problems of length $n = 10$, we test both models on sequences of length $n = 10$ and $n = 30$. For a sequence of length $n$ both models predict values not only for the entire expression but also for all $n$ subproblems where index $n$ corresponds to evaluating the entire sequence. For sequences of both lengths, we evaluate accuracy for the predicted answers on all subproblems. Results are shown in Figure 3.

## 5.2 DISCUSSION

We argue that the LSTM fails on the RPN task. This is not immediately obvious: from Figure 3, both the small and large LSTM baselines approximately match our model's performance on the first 5 subproblems of the $n = 10$ dataset. From $n = 6$ to $n = 10$, the performance gap grows between our models – the small LSTM is unable to learn deep stack behavior, and performance decays sharply.

The $n = 30$ dataset reveals the failure. The LSTM's performance is far worse on the first few subproblems of this dataset than on the test set of the original task. This is not an error: recall the question formatting [NUM]*$(n + 1)$-[OP]*$n$. The leading subproblems do not correspond to the leading tokens of the question, but rather to a central crop. For example, the first two subproblems of "12345+-*/" are given by "345+-", not "12345" – the latter is not a valid expression. The rapid increase in error on the LSTM implies that it did not learn this property, let alone the stack structure. Instead, it memorized all possible subproblems of length $n \in \{1, 2, 3\}$ expressions preceding the first few OP tokens. Performance quickly decays to L1 error greater than 2.0, which corresponds to mostly noise (the standard deviation of answers minus the first few subproblems is approximately 6.0). In contrast, our model's explicit incorporation of the stack assumption results in a smooth generalization curve with a gradual decay in performance as problem length increases.

We briefly address a few likely concerns with our reasoning. First, one might argue that DDRstack cannot be compared to an LSTM, as the latter does not incorporate explicit knowledge of the problem structure. While this evaluation is correct, it is antithetical to the purpose of our architecture. The LSTM baseline does not incorporate this additional information because there is no obvious way to include it – the prevailing approach would be to ignore it and then argue the model's superiority on the basis that it performs well with less supervision. This logic might suggest implicit reasoning approaches such as StackRNN, which attempt to model the underlying data structure without direct supervision. However, we do not expect such approaches to scale to RPN: the hardest task on which StackRNN was evaluated is binary addition. While StackRNN exhibited significantly better generalization compared to the LSTM baseline, the latter did not completely fail the task. In contrast, RPN is a more complex task that completely breaks the baseline LSTM. While we did not evaluate Stack-RNN on RPN (the original implementation is not compatible modern frameworks), we consider it highly improbably that StackRNN would generalize to RPN, which was intentionally designed to be difficult without additional supervision. In contrast, our dynamic approach achieves a dramatic increase in performance and generalization precisely by efficiently incorporating additional supervision. StackRNN is to DDRstack as FiLM is to DDRprog: one motive is to maximize performance with minimal supervision whereas our motive is to leverage structural data to solve harder tasks.

## 6 CONCLUSION

The DDR framework facilitates high level reasoning in neural architectures by enabling networks to leverage additional structural information. Our approach resolves differentiability issues common in interfering with discrete logical data and is easily adapted to the specific of each problem. Our work represents a clean synthesis of the modeling capabilities of IEP/NMN and NPI through a differentiable forking mechanism. We have demonstrated efficacy through two applications of our framework. DDRprog achieves a moderate improvement over previous state-of-the-art on CLEVR with greatly increased consistency and reduced model size. DDRstack succeeds on RPN where a much larger baseline LSTM fails to attain generalization. It is our intent to continue refining the versatility of our architecture, including more accurate modeling of the fork module, as mentioned in our CLEVR VQA discussion. Our architecture and its design principles enable modeling of complex data structure assumptions across a wide class of problems where standard monolithic approaches would ignore such useful properties. We hope that this increase in interoperability between discrete data structures and deep learning architectures aids in motivating higher level tasks for the continued progression and development of neural reasoning.

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

APPENDIX

Table 2: Architectural details of subnetworks in DDRprog as referenced in Figure 1 and Algorithm 1. Finegrained layer details are provided in tables 4-8. Source will be released pending publication.

| Subnetwork | Details |
|---|---|
| ResNetFeaturizer | Features from ResNet101 pretrained on ImageNet, as in IEP and FiLM |
| LSTM | 2-Layer LSTM that encodes the question |
| CNN | IEP classifier variant; produces a flat visual state. |
| Controller | Recurrent Highway Network for language and CNN Encoded visual states |
| Cells | IEP set of unary and binary modules, plus our fork module and pads |

Table 3: Hyperparameter details for DDRprog. Only the learning rate and model size were coarsely cross validated due to hardware limitations: hyperparameter are not optimal.

| Module | Architecture |
|---|---|
| Hidden dimension, all convolutional modules | 64 |
| Hidden dimension, all recurrent modules | 128 |
| Question encoder depth | 2 |
| Recurrent controller depth | 3 |
| Question vocabulary embedding dimension | 300 |
| Learning rate | 1e-4 |

Table 4: ResNetFeaturizer

| Index | Layer | Output Size |
|---|---|---|
| (1) | Input Image | $3 \times 224 \times 224$ |
| (2) | ResNet101 conv4_6 | $1024 \times 14 \times 14$ |
| (3) | Conv($3 \times 3, 1024 \to h$) | $h \times 14 \times 14$ |
| (4) | ReLU | $h \times 14 \times 14$ |
| (5) | Conv($3 \times 3, h \to h$) | $h \times 14 \times 14$ |
| (6) | ReLU | $h \times 14 \times 14$ |

Table 5: Unary Module

| Index | Layer | Output Size |
|-------|-------|-------------|
| (1) | Previous Module Output | h $\times$ 14 $\times$ 14 |
| (2) | Conv(3 $\times$ 3, $h \rightarrow h$) | h $\times$ 14 $\times$ 14 |
| (3) | ReLU | h $\times$ 14 $\times$ 14 |
| (4) | Conv(3 $\times$ 3, $h \rightarrow h$) | h $\times$ 14 $\times$ 14 |
| (5) | Residual: Add (1) and (4) | h $\times$ 14 $\times$ 14 |
| (6) | ReLU | h $\times$ 14 $\times$ 14 |
| (7) | InstanceNorm | h $\times$ 14 $\times$ 14 |

Table 6: Binary Module

| Index | Layer | Output Size |
|-------|-------|-------------|
| (1) | Previous Module Output | h $\times$ 14 $\times$ 14 |
| (2) | Previous Module Output | h $\times$ 14 $\times$ 14 |
| (3) | Concatenate (1) and (2) | 2h $\times$ 14 $\times$ 14 |
| (4) | Conv(1 $\times$ 1, $2h \rightarrow h$) | h $\times$ 14 $\times$ 14 |
| (5) | ReLU | h $\times$ 14 $\times$ 14 |
| (6) | Conv(3 $\times$ 3, $h \rightarrow h$) | h $\times$ 14 $\times$ 14 |
| (7) | ReLU | h $\times$ 14 $\times$ 14 |
| (8) | Conv(3 $\times$ 3, $h \rightarrow h$) | h $\times$ 14 $\times$ 14 |
| (9) | Add (5) and (8) | h $\times$ 14 $\times$ 14 |
| (10) | ReLU | h $\times$ 14 $\times$ 14 |

Table 7: Fork Module

| Index | Layer | Output Size |
|-------|-------|-------------|
| (1) | Previous Module Output | h $\times$ 14 $\times$ 14 |
| (2) | Previous Module Output | h $\times$ 14 $\times$ 14 |
| (3) | Concatenate (1) and (2) | 2h $\times$ 14 $\times$ 14 |
| (4) | Conv(1 $\times$ 1, $2h \rightarrow 6h$) | 6h $\times$ 14 $\times$ 14 |
| (5) | ReLU | 6h $\times$ 14 $\times$ 14 |
| (6) | Conv(3 $\times$ 3, $6h \rightarrow 6h$) | 6h $\times$ 14 $\times$ 14 |
| (7) | ReLU | 6h $\times$ 14 $\times$ 14 |
| (8) | Conv(3 $\times$ 3, $6h \rightarrow 6h$) | 6h $\times$ 14 $\times$ 14 |
| (9) | Add (5) and (8) | 6h $\times$ 14 $\times$ 14 |
| (10) | ReLU | 6h $\times$ 14 $\times$ 14 |
| (11) | Conv(1 $\times$ 1, $6h \rightarrow h$) | h $\times$ 14 $\times$ 14 |

Table 8: CNN

| Index | Layer | Output Size |
|-------|-------|-------------|
| (1) | Previous Module Output | h $\times$ 14 $\times$ 14 |
| (2) | Conv(3 $\times$ 3, $h \rightarrow h$) | h $\times$ 14 $\times$ 14 |
| (3) | ReLU | h $\times$ 14 $\times$ 14 |
| (4) | Conv(3 $\times$ 3, $h \rightarrow h$) | h $\times$ 14 $\times$ 14 |
| (5) | Residual: Add (1) and (4) | h $\times$ 14 $\times$ 14 |
| (6) | ReLU | h $\times$ 14 $\times$ 14 |
| (7) | MaxPool(2 $\times$ 2, $h \rightarrow h$) | h $\times$ 7 $\times$ 7 |
| (8) | Conv(3 $\times$ 3, $h \rightarrow \frac{1}{2}h$) | $\frac{1}{2}$ h $\times$ 5 $\times$ 5 |
| (9) | Flatten | $\frac{1}{2}$h*5*5 |
| (10) | Linear($\frac{1}{2}$h*5*5 $\times$ 1024) | 1024 |
| (11) | ReLU | 1024 |
| (12) | Linear(1024 $\times$ Classes) | Classes |

Table 9: Success examples on CLEVR. The numerical prefix on each program function is its arity.

- Image Index: 5156
- Question: there is a small purple rubber object; what shape is it ?
- Program (label): 1_filter_size_small 1_filter_color_purple 1_filter_material_rubber 1_unique 1_query_shape
- Answer (predicted, label): cylinder, cylinder

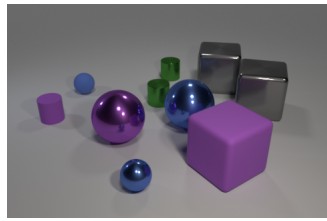

- Image Index: 2364
- Question: what number of tiny brown shiny things are to the right of the matte sphere that is on the left side of the tiny red object to the right of the small yellow object ?
- Program (label): 1_filter_size_small 1_filter_color_yellow 1_unique 1_relate_right 1_filter_size_small 1_filter_color_red 1_unique 1_relate_left 1_filter_material_rubber 1_filter_shape_sphere 1_unique 1_relate_right 1_filter_size_small 1_filter_color_brown 1_filter_material_metal 1_count
- Answer (predicted, label): 1, 1

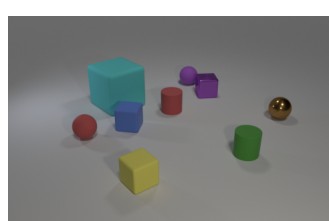

- Image Index: 4287
- Question: there is a block that is behind the cyan metal thing; what material is it ?
- Program (label): 1_filter_color_cyan 1_filter_material_metal 1_unique 1_relate_behind 1_filter_shape_cube 1_unique 1_query_material
- Answer (predicted, label): rubber, rubber

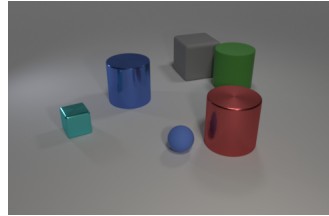

- Image Index: 10332
- Question: is the number of tiny gray matte cubes that are behind the tiny gray matte object the same as the number of cyan shiny cylinders ?
- Program (label): 1_filter_color_cyan 1_filter_material_metal 1_filter_shape_cylinder 1_count 0_fork 1_filter_size_small 1_filter_color_gray 1_filter_material_rubber 1_unique 1_relate_behind 1_filter_size_small 1_filter_color_gray 1_filter_material_rubber 1_filter_shape_cube 1_count 2_equal_integer
- Answer (predicted, label): yes, yes

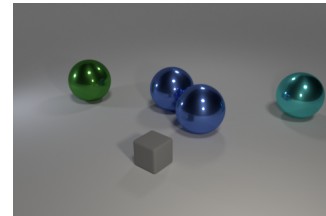

Table 10: Failure examples on CLEVR. The numerical prefix on each program function is its arity. Many errors are the result of occlusions. This can be extreme: the second error example is only answerable by process of elimination.

- Image Index: 6688
- Question: what is the color of the tiny matte cylinder ?
- Program (label): 1_filter_size_small 1_filter_material_rubber 1_filter_shape_cylinder 1_unique 1_query_color
- Answer (predicted, label): brown, gray

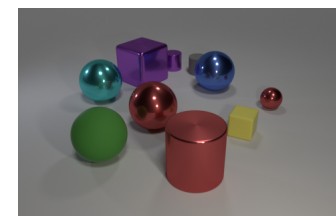

- Image Index: 8307
- Question: there is a small cube that is made of the same material as the gray object; what is its color ?
- Program (label): 1_filter_color_gray 1_unique 1_same_material 1_filter_size_small 1_filter_shape_cube 1_unique 1_query_color
- Answer (predicted, label): purple, yellow

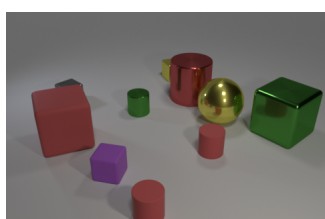

- Image Index: 1902
- Question: what color is the block that is to the left of the big yellow matte block and behind the large blue shiny block ?
- Program (label): 1_filter_size_large 1_filter_color_yellow 1_filter_material_rubber 1_filter_shape_cube 1_unique 1_relate_left 0_fork 1_filter_size_large 1_filter_color_blue 1_filter_material_metal 1_filter_shape_cube 1_unique 1_relate_behind 2_intersect 1_filter_shape_cube 1_unique 1_query_color
- Answer (predicted, label): green, blue

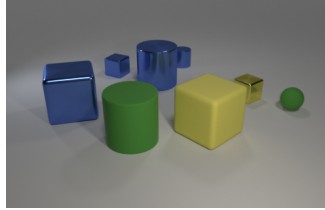

- Image Index: 8543
- Question: are there fewer small purple rubber things that are behind the green metallic cylinder than small things that are in front of the tiny matte block ?
- Program (label): 1_filter_size_small 1_filter_material_rubber 1_filter_shape_cube 1_unique 1_relate_front 1_filter_size_small 1_count 0_fork 1_filter_color_green 1_filter_material_metal 1_filter_shape_cylinder 1_unique 1_relate_behind 1_filter_size_small 1_filter_color_purple 1_filter_material_rubber 1_count 2_less_than
- Answer (predicted, label): no, yes

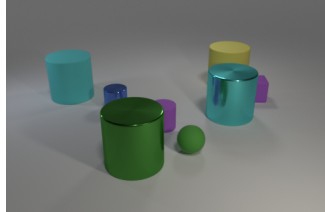

