# OpenReview forum: "DDRprog: A CLEVR Differentiable Dynamic Reasoning Programmer"
_ICLR.cc/2018/Conference — Reject_

### Official Review · AnonReviewer2 · 2017-11-28
**Good performance but requires explicit program supervision to slightly beat the state of the art**

**Rating:** 5
**Confidence:** 2

**Review:**


Summary: This paper leverages an explicit program format and proposes a stack based RNN to solve question answering. The paper shows state-of-the art performance on the CLEVR dataset.

Clarity:
- The description of the model is vague: I have to looking into appendix on what are the Cell and Controller function.
- The authors should also improve the intro and related work section. Currently there is a sudden jump between deep learning and the problem of interest. Need to expand the related work section to go over more literature on structured RNN.

Pros:
- The model is fairly easy to understand and it achieves state-of-the-art performance on CLEVR.
- The model fuses text and image features in a single model.

Cons:
- This paper doesn’t mention recursive NN (Socher et al., 2011) and Tree RNN (Tai et al., 2015). I think they have fairly similar structure, at least conceptually, the stack RNN can be thought as a tree parser. And since the push/pop operations are static (based on the inputs), it’s no more different than encoding the question structure in the tree edges.
- The IEP (Cells) module (Johnson et al., 2017) seems to do all the heavy-lifting in my opinion. That’s why the proposed method only uses 9M parameters. The task isn’t very challenging to learn because all the stack operations are already given. Table 1 should note clearly which methods use problem specific parsing information to train and which use raw text. Based on my understanding of FiLM at least, they use raw words instead of groundtruth parse trees. So it’s not very surprising that the proposed method can outperform FiLM (by a little bit).
- I don’t fully agree with the title - the stack operations are not differentiable. So whatever network that outputs the stack operation cannot be jointly learned with gradients. This is based on the if-else statements I see in Algorithm 1.

Conclusion:
- Since the novelty is limited and it requires explicit program supervision, and the performance is only on par with the state-of-the-art (FiLM), I am not convinced that this paper brings enough contribution to be accepted. Weak reject.

References:
- Socher, R., Lin, C., Ng, A.Y., Manning, C.D. Parsing Natural Scenes and Natural Language with Recursive Neural Networks. The 28th International Conference on Machine Learning (ICML 2011).
- Tai, K.S., Socher, R., Manning C.D. Improved Semantic Representations From Tree-Structured Long Short-Term Memory Networks. The 53rd Annual Meeting of the Association for Computational Linguistics (ACL 2015).

---

> ### Author Response · Authors · 2018-01-05
> **Updated related works, clarified model, consistency of performance, and ~4X reduction in error on an important subtask**
>
>
> We are sorry to hear that you are not convinced of the merit of our work. Your concerns seem to stem from a lack of clarity. This was a common theme--all three reviewers emphasize a lack of clarity overall. We did our best to address this in our initial submission—to that end, we included a visual, tabular, and algorithmic representation of DDRprog with a full page description and similar information for DDRstack. However, in the process of emphasizing algorithmic details, we neglected proper treatment of the extensive motivation for CLEVR over conventional VQA datasets and assumed far too much domain familiarity. The largest effort of our revision has been to increase accessibility on this front. We have reworked the intro and related works section dramatically to better capture the motivation behind CLEVR, the motivation behind our framework, and concretely how the details of our architectures combine to address all motives. We particularly emphasize the novelty of our approach as a general reasoning framework, which was not clear from our initial submission.
>
> We have reviewed and reconsidered additional structured RNN literature. We now make mention of recursive NN and Tree RNN and detail the difference from our present work. The push/pop operations are not static in our CLEVR architecture—they are given as supervision at train time, but are learned and coupled to module prediction at test time. In our RPN architecture, the order of push/pop operations is static, but our module reuse scheme would be difficult to motivate from the static context of e.g. Tree RNN works—we use the RPN task to demonstrate the flexibility of our framework across differing degrees of supervision.
>
> You are correct that the IEP/NMN cells do the heavy lifting—but this is precisely our intent. Our network structure overall is completely different from IEP despite reusing many of that work’s subnetworks out of convenience. In particular, our network not only has the ability to predict and execute modules one at a time (IEP must predict and compile a static program for the entire question), but it also observes the outputs of module executions and uses this to predict the next module. This is a significant contribution of our work—while IEP relies on a larger variant of the same module structure, we achieve a 2X reduction in relative error over IEP on all unary tasks with a much smaller model.
>
> While we are largely concerned with building a general framework for discrete, high-level reasoning rather than raw accuracy on CLEVR, our performance increment over FiLM is actually quite significant. In particular, FiLM suffers from a clear logical deficiency on Compare Integer questions—in this case, we achieve a roughly 4X relative improvement in accuracy from 93.8 to 98.4 percent. Furthermore, our architecture is the only model proposed thus far that exhibits strong, consistent performance across all tasks—all other models exhibit inconsistently poor performance on at least one subtask relative to performance on the task as a whole. We have dramatically clarified this additional merit of our model in the Discussion section.
>
> You are correct that the stack operations themselves are not differentiable. However, the prediction of these operations is learned differentiably: our architecture learns to fork subprocesses, and the push/pop behavior is part of the forking behavior. There is one significant non-differentiable aspect of our network: the pathway from the answer loss to the program cell loss is not directly learnable. However, there is an important indirect interaction between the losses, which is a key contribution of our architecture—unlike in IEP, we maintain a visual state that is directly used by the controller during module prediction, and this visual state is affected by the question answer gradients. This it the main difference between our model and IEP--the proof of the utility of this mechanism is our model’s performance gain vs. IEP despite significantly smaller model size.

---

> > ### Author Response · Authors · 2018-01-05
> > **Clarification of the importance of RPN**
> >
> >
> > Though you did not mention our RPN experiments, this task is crucial to the motivation of our work. Perhaps CLEVR is already too easy for modern architectures, but not every task is feasible without strong supervision. This is the purpose of our framework: generality and flexibility in incorporating complex annotation information. To this end, we present an expression evaluation task that completely breaks a standard LSTM. In contrast, our framework leverages a simple assumption to yield strong performance with a 10X smaller network.
> >
> > In closing: DDRprog uses increased supervision to achieve state-of-the-art performance with a small network. We achieves comparable performance to FiLM across 3/5 subtasks and much stronger performance on Count and especially Compare Integer tasks. DDRprog is the only architecture to date that does not exhibit particularly poor performance on at least one subtask. We present DDRstack as a successful application of the same dynamic, increased-supervision framework to the RPN task where a 10X larger LSTM fails to generalize. DDRprog and DDRstack represent applications of our general reasoning framework--a novel approach to learning discrete structural data with increased differentiability and generality over all prior approaches.

---

### Official Review · AnonReviewer3 · 2017-12-01
**Presentation of the approach is not clear**

**Rating:** 5
**Confidence:** 2

**Review:**

Summary:
The paper presents a generic dynamic architecture for CLEVR VQA and Reverse Polish notation problems. Experiments on CLEVR show that the proposed model DDRprog outperforms existing models, but it requires explicit program supervision. The proposed architecture for RPN, called DDRstack outperforms an LSTM baseline.

Strengths:
— For CLEVR VQA task, the proposed model outperforms the state-of-the-art with significantly less number of parameters.
— For RPN task, the proposed model outperforms baseline LSTM model by a large margin.

Weaknesses:
— The paper doesn’t describe the model clearly. After reading the paper, it’s not clear to me what the components of the model are, what each of them take as input and produce as output, what these modules do and how they are combined. I would recommend to restructure the paper to clearly mention each of the components, describe them individually and then explain how they are combined for both cases - DDRprog and DDRstack.
— Is the “fork” module the main contribution of the paper? If so, at least this should be described in detail. So, if no fork module is required for a question, the model architecture is effectively same as IEP?
— Machine accuracy is already par with human accuracy on CLEVR and very close to 100%. Why is this problem still important?
— Given that the performance of state-on-art on CLEVR dataset is already very high ( <5% error) and the performance numbers of the proposed model are not very far from the previous models, it is very important to report the variance in accuracies along with the mean accuracies to determine if the performance of the proposed model is statistically significantly better than the previous models or not.
— In Figure 4, why are the LSTM32/128 curves different for Length 10 and Length 30 till subproblem index 10? They are both trained on the same training data, only test data is of different length and ideally both models should achieve similar accuracy for the first 10 subproblems (same trend as DDRstack).
— Why is DDRstack not compared to StackRNN?
— Can the authors provide training time comparison of their model and other/baseline models? Because that is more important than the number of epochs required in training.
— There are only 3 curves (instead of 4) in Figure 3.
— In a number of places, the authors are referring to left and right program branches. What are they? These names have not being defined formally in the paper.

Overall:
I think the research work in the paper is interesting and significant, but given the current presentation and level of detail in the paper, I don’t think it will be helpful for the research community. By proper restructuring of paper and adding more details, the paper can be converted to a solid submission.

---

> ### Author Response · Authors · 2018-01-05
> **Presentation of approach clarified**
>
> We agree with your overall evaluation—from your commentary, it is clear that the better part of our work’s contribution is currently obscured by unclear presentation. This was a common theme--all three reviewers emphasize a lack of clarity overall. We did our best to address this in our initial submission—to that end, we included a visual, tabular, and algorithmic representation of DDRprog with a full page description and similar information for DDRstack. However, in the process of emphasizing algorithmic details, we neglected proper treatment of the extensive motivation for CLEVR over conventional VQA datasets and assumed far too much domain familiarity.
>
> The largest effort of our revision has been to clarify our presentation and increase accessibility. We have reworked the intro and related works section dramatically to better capture the motivation behind CLEVR, the motivation behind our framework, and concretely how the details of our architectures combine to address all motives. We also include a thorough appendix of architecture tables sufficient to reproduce our result. We will also open source the entire project pending publication.
>
> Discounting the fork module, our architecture still represents a significant improvement over IEP. On unary programs that do not require a fork module, we attain a 2X improvement in relative error over IEP with a 4X smaller model. While we do include the fork architecture in our revision, the layer details are less relevant than stack behavior that forking enables. In our revision, we particularly emphasize the novelty of our approach as a general reasoning framework, which was not clear from our initial submission. Furthermore, while IEP predicts programs dynamically, the actual execution is static. In contrast, our architecture is fully dynamic, predicting and executing modules on the fly. Previous module outputs are used to update an internal visual state, which is in turn used to predict the next module. This positions our architecture as a superset of IEP/NMN and NPI capable of jointly learning functional modules and complex stack traces.
>
> You are correct that CLEVR is effectively solved. There are two reasons that the task is still important. First, every competitive previous work exhibits a deficiency in at least one category of reasoning: all prior works obtain curiously poor performance on at least one important CLEVR subtask. As the first approach to attain close consistency across all tasks, our work solves this problem. Second, CLEVR remains the best proxy task for high-level visual reasoning because of its discrete program annotations. This is more in line with the purpose of our work as a general framework for differentiably leveraging logical annotations. However, we should note that our performance numbers do in fact represent a significant improvement over prior approaches: as mentioned above, we achieve a 2X relative improvement over IEP on Count, Exist, and Query tasks with a much smaller model and also maintain comparable results on the two Compare tasks. FiLM suffers from a clear logical deficiency on Compare Integer questions—in this case, we achieve a roughly 4X relative improvement in accuracy from 93.8 to 98.4 percent. We also achieve a significant improvement from 94.5 to 96.5 percent on Count Questions; performance is comparable on all other subtasks.
>
> The discrepancy in Figure 4 is a critical point to our argument addressed in depth in the RPN discussion section. The first 10 subproblems on the length 30 task correspond to a center crop of the data sequence: the LSTM is confused by the additional leading 20 numbers in the sequence. This indicates that LSTM has not learned the stack structure and completely fails the task. This result is crucial to the motivation of our work. Perhaps CLEVR is already too easy for modern architectures, but not every task is feasible without strong supervision. This is the purpose of our framework: generality and flexibility in incorporating complex annotation information. To this end, we present an expression evaluation task that completely breaks a standard LSTM. In contrast, our framework leverages a simple assumption to yield strong performance with a 10X smaller network.

---

> > ### Author Response · Authors · 2018-01-05
> > **continued**
> >
> >
> > Finally, we address the rationale behind several smaller objections:
> > - On the topic of variance, the CLEVR test set contains 150K and is very consistent with the validation set according to the CLEVR authors--we ran the test set only once on our best model. If your concern stems from the reported 0.4% standard deviation reported by the authors of FiLM, it appears from their open sourced release that they trained for a fixed number of iterations and also reported that their model did not necessarily converge.
> > - There are in fact 4 cures in Figure 3, but the train and validation curves for our model overlap due to a lack of overfitting; we have added this to the caption.
> > - Training time is non-standard for dynamic architectures because it is often not meaningful. Many dynamic architectures currently run more quickly on CPU than GPU simply because research in this area has far outstripped framework optimization for these models. By FLOPs, our model is more efficient than all architectures except RN. However, RN is extremely slow to train—the authors used 10 GPUs in a distributed setting whereas all other models run on single cards. As a rough idea of computational footprint, our research was conducted with two personally owned cards.
> > - The left/right program branch notation was admittedly confusing and has been stricken from our work. We were originally referring to the fact that binary programs can be written vertically as a tree with two program branches; we have reworded this in our discussions for clarity.
> > - We did discuss StackRNN, but they are not comparable architectures: StackRNN does not leverage additional annotations to learn the stack structure. While this may seem more general, the original work strongly suggests that StackRNN is limited to simpler problems. In contrast, our architecture makes strong structural assumptions to solve the much harder RPN task. This has been expanded upon and clarified in our revision.
> >
> > In closing: DDRprog uses increased supervision to achieve state-of-the-art performance with a small network. We achieves comparable performance to FiLM across 3/5 subtasks and much stronger performance on Count and especially Compare Integer tasks. DDRprog is the only architecture to date that does not exhibit particularly poor performance on at least one subtask. We present DDRstack as a successful application of the same dynamic, increased-supervision framework to the RPN task where a 10X larger LSTM fails to generalize. DDRprog and DDRstack represent applications of our general reasoning framework--a novel approach to learning discrete structural data with increased differentiability and generality over all prior approaches.

---

### Official Review · AnonReviewer1 · 2017-12-03
**Motivation behind the proposed model needs improvement, paper writing about model architecture not easy to follow.**

**Rating:** 6
**Confidence:** 2

**Review:**

Summary:
The paper proposes a novel model architecture for the visual question answering task in the CLEVR dataset. The main novelty of the proposed model lies in its problem specific differentiable forking mechanism that is designed to encode complex assumptions about the data structure in the given problem. The proposed model is also applied on the task of solving Reverse Polish Notation (RPN) expression. On the CLEVR dataset, the proposed model beats the state of the art by 0.7% with ~5 times fever learning parameters and in about ~1/2 as many epochs. For the RPN task, the proposed model beats an LSTM baseline by 0.07 in terms of L1 error with ~10 times fewer parameters.

Strengths:
1.	The proposed model is novel and interesting.
2.	The performance of the proposed model on the “Count” questions in the CLEVR dataset is especially better than existing models and is worth noting.
3.	The discussion on the tradeoff between tacking difficult problems and using the knowledge of program structure is engaging.

Weaknesses:
1.	The paper writing about the model architecture can be improved. As of now, it is not easy to follow.
2.	The motivation behind the proposed model for the CLEVR task has been explained example of one type of questions – “How many objects are red or spheres?”. It is not clear how the proposed model is better than existing models (in terms of model architecture) for other types of questions.
3.	The motivation behind the proposed model for the RPN task is not strong enough. It is not clear why is machine learning needed for the RPN task? Is the idea that we do not want to use domain knowledge about which symbols correspond to operations vs. which correspond to numbers? Or is there more to it?
4.	The proposed model needs to use the information about program structure. It would be good if authors could comment on how can the proposed model be used to answer natural language questions about images, such as, those in the VQA dataset (Antol et al., ICCV 2015).
5.	The paper does not have any qualitative examples for either of the two tasks. Qualitative examples of successes and failures would be helpful to better position proposed model against existing models.

Overall: The experimental results look good, however, the proposed model needs to be better motivated. The paper writing, especially the “DDR Architecture” section needs improvement to make it easy to follow. A discussion on how the existing model can be adapted for natural language questions would be good to add.

---

> ### Author Response · Authors · 2018-01-05
> **Improved motivation and paper writing, especially in the Introduction and Architecture sections**
>
> Thank you for your commentary despite the admittedly unclear writing in our initial submission. We address each of your concerns below:
>
> 1. We did our best present clean visual, tabular, and algorithmic representations of DDRprog and similar information for DDRstack. However, we neglected proper treatment of the extensive motivation for CLEVR over conventional VQA datasets and assumed far too much domain familiarity. The largest effort of our revision has been to increase accessibility on this front. We have completely reframed the motivation behind CLEVR, the motivation behind our architecture, and concretely how the details of our architecture combine to address all motives.
>
> 2. While IEP predicts programs dynamically, the actual execution is static. In contrast, our architecture is fully dynamic, predicting and executing modules on the fly. Previous module outputs are used to update an internal visual state, which is in turn used to predict the next module. This feedback loop enabled by our dynamic framework is the motivation behind DDRprog and also the reason for its success. This creates a fundamentally different behavior from IEP, even on unary programs that do not require a fork module: on unary subtasks, we attain a 2X improvement in relative error over IEP with a 4X smaller model. The fork module itself allows our architecture to model generic trees: this positions our architecture as a superset of IEP/NMN and NPI capable of jointly learning functional modules and complex stack traces. Our architecture is built specifically for high level reasoning: it is the only model with consistently high performance across all subtasks. FiLM suffers from a clear logical deficiency on Compare Integer questions—in this case, we achieve a roughly 4X relative improvement in accuracy from 93.8 to 98.4 percent. We also achieve a significant improvement from 94.5 to 96.5 percent on Count Questions; performance is comparable on all other subtasks.
>
> 3. The RPN task is itself the motivating example for our work. DDRstack does not simply outperform the LSTM baseline on RPN by 0.07: it succeeds where the LSTM fails entirely. The RPN task is a toy problem that turns out to be very hard without additional supervision: the purpose of the expression evaluation task is that it fundamentally involves a parse tree—a stack based algorithm. The discrepancy between length 10 and length 30 performance in Figure 4 is critical here. The first 10 subproblems on the length 30 task correspond to a center crop of the data sequence (e.g. the first subproblem of “12345+*-/” is “45+”) : the LSTM is confused by the additional leading 20 numbers in the sequence. This indicates that LSTM has not learned the stack structure and completely fails the task. In contrast, DDRstack leverages a simple assumption to yield strong performance with a 10X smaller network. This is the purpose of our framework: generality and flexibility in incorporating complex annotation information. Perhaps CLEVR is already too easy and solvable by static architectures such as FiLM, but not every task is feasible without strong supervision.
>
> 4. Following the line of reasoning above, the future direction of this work is more general reasoning over knowledge graphs. We prefer synthetic tasks for ease of annotation generation, but plan on transitioning to natural image tasks in the long term--Perhaps not Antol et al.’s VQA dataset, but the natural language and image scene graph dataset Visual Genome.
>
> 5. Good point! Appendix included.
>
> In closing: DDRprog uses increased supervision to achieve state-of-the-art performance with a small network. We achieves comparable performance to FiLM across 3/5 subtasks and much stronger performance on Count and especially Compare Integer tasks. DDRprog is the only architecture to date that does not exhibit particularly poor performance on at least one subtask. We present DDRstack as a successful application of the same dynamic, increased-supervision framework to the RPN task where a 10X larger LSTM fails to generalize. DDRprog and DDRstack represent applications of our general reasoning framework--a novel approach to learning discrete structural data with increased differentiability and generality over all prior approaches.

---

### Author Response · Authors · 2018-01-05
**Thanks To All Reviewers**

We thank all reviewers for their commentary. We have largely restructured and streamlined the paper--particularly the introduction and architecture section--and have taken into account all reviewer specific commentary in the new version. We hope that we have assuaged your concerns and adequately clarified points of confusion through significant improvement of the writing. Please see reviewer specific responses below.

---

### Decision · Program_Chairs · 2018-01-29
**ICLR 2018 Conference Acceptance Decision**

**Decision:**

Reject

**Comment:**

The reviewers generally agree that the DDRprog method is both novel and interesting, while also seeing merit in outperformance of related methods in the empirical results. However, There were a lot of complaints about the writing quality, the clarity of the exposition, and unclear motivation of some of the work.  The reviewers also noted insufficient comparisons and discussions regarding relevant prior art, including recursive NNs, Tree RNNs, IEP, etc.  While the authors have made substantial revisions to the manuscript, with several additional pages of exposition, reviewers have not raised their scores or confidence in response.